# LEARNING TO CLUSTER

**Benjamin B. Meier, Thilo Stadelmann & Oliver Dürr**
ZHAW Datalab, Zurich University of Applied Sciences
Winterthur, Switzerland
`benjamin.meier70@gmail.com, stdm@zhaw.ch, oliver.duerr@gmail.com`

## ABSTRACT

We propose a novel neural network architecture to learn the task of clustering end-to-end: salient features for any similarity criterion specified through weakly labeled training data are extracted with an embedding network; during evaluation, the network groups similar data of any modality together, by assigning a probabalistic cluster index, and further gives a probabilistic estimate of the number of clusters. The method is evaluated on 2D point data, speaker data from the TIMIT corpus, and images from the COIL-100 dataset, reaching promising results.

## 1 INTRODUCTION

Classic clustering algorithms like k-Means (MacQueen et al., 1967), EM (Dempster et al., 1977), or DBSCAN (Ester et al., 1996) depend crucially on an implicit measure of the semantic similarity between examples. However, for modalities like images (Liu et al., 2007), speakers (Stadelmann & Freisleben, 2009), or text (Aggarwal & Zhai, 2012), the similarity is not well known or ambiguous (even for 2D points, one has to know if clusters are defined by density or distance to a centroid).

Recently, clustering of high-dimensional data with complex similarity structure has been approached with good success using neural nets to extract well-suited features for clustering called embeddings, which are then subject to a subsequent offline clustering process (Mikolov et al., 2013; Schroff et al., 2015; Lukic et al., 2017). However, training to create embeddings is done on a surrogate task (e.g., classification) instead of being optimizable and applicable end-to-end as done in our method.

Related work by Makhzani et al. (2015) goes beyond mere metric learning (Xing et al., 2003; Hoffer & Ailon, 2015) and finally predicts the cluster index for specific groups of data seen during training. Instead, our method does neither require that the training data contains any of the clusters expected during evaluation, nor to know the real number of clusters $k$. Yang et al. (2016) present an end-to-end neural clustering approach and evaluate it on COIL-100 (Nayar et al., 1996). They interpret agglomerative hierarchical clustering as a recurrent process that is optimizable through the complete network. However, our method generalizes to different modalities besides images; moreover, it is not bound to a static predefined clustering scheme, but learns the complete "algorithm".

Our contribution in this paper is a method able to learn salient features *and* to group examples, by means of a novel neural net architecture. The network input is a set of examples, the output a probabalistic clustering (including the predicted number of clusters $k$). It is trained in a supervised fashion using weakly labeled data, and afterwards applicable to unlabeled data from the same distribution, but different and unknown numbers of clusters. For example, we train on a specific set of voices, and use the resulting model to cluster completely different sets of speakers. We evaluate our approach on different datasets, showing promising performance and a high degree of generality for data types from 2D points to audio snippets and images.

## 2 A MODEL FOR END-TO-END CLUSTERING OF ARBITRARY EXAMPLES

Our method learns to cluster end-to-end purely ab initio, without the need to explicitly specify a notion of similarity, only providing the information whether two examples belong together. It uses as input $n \geq 1$ examples $x_i$, where $n$ may be different during training and evaluation. The network's output is two-fold: a distribution $P(k)$ over the cluster count $k$; and a distribution $P(\cdot \mid x_i, k)$ over the cluster index for each input example $x_i$ and for each $k$. The network architecture (see

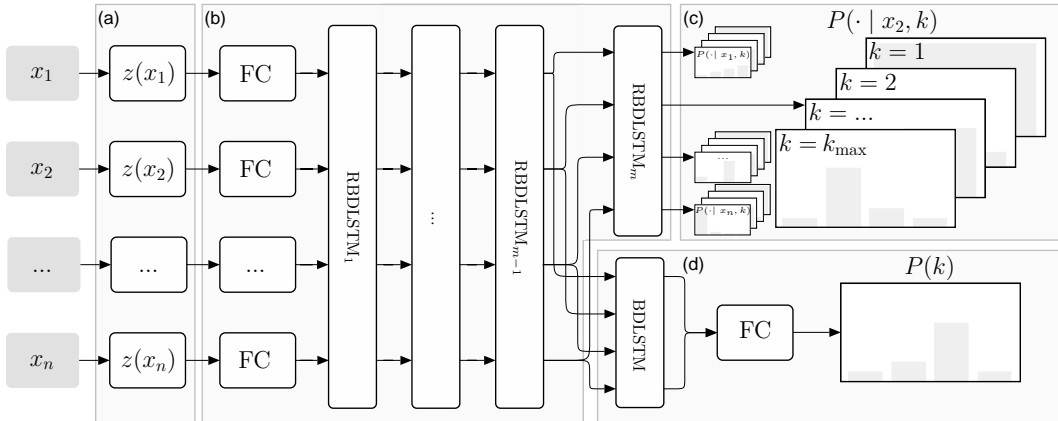

Figure 1: Our complete model, consisting of (a) the embedding network, (b) clustering network, (c) cluster-assignment network and (d) cluster-count estimating network.

Fig. 1) allows the flexible use of different input types, e.g. images, audio or 2D points. An input $x_i$ is first processed by an embedding network (a) that produces a lower-dimensional representation $z_i = z(x_i)$. The dimension of $z_i$ may vary depending on the used data type. For example, 2D points do not require any embedding network. A fully connected layer (FC) at the beginning of the clustering network (b) with 288 units and LeakyReLU activation ($\alpha = 0.3$) is then used to bring all embeddings to the same size. The goal of this sub-network is to compare each input with all others, in order to suggest a grouping. To process an arbitrary number of examples $n$, we use stacked ($m = 14$) residual bi-directional versions (RBDLSTM) of the cells described by Wu et al. (2016).

The first of two outputs is modeled by the cluster assignment network (c). It contains a softmax-layer $P(\ell \mid x_i, k)$ which assigns a cluster index $\ell$ to each input $x_i$, given $k$ clusters (i.e., we get a distribution over possible cluster assignments for each input and every possible number of clusters). The second output, produced by the cluster-count estimating network (d), is built from one BDLSTM-layer with 128 units. We concatenate its first and the last output vector into a fully connected layer (256 units) using again LeakyReLUs ($\alpha = 0.3$). The subsequent softmax-activation models the distribution $P(k)$ ($1 \leq k \leq k_{\max}$). Given $P(k)$, the most probable cluster of the cluster-assignment network can be chosen for each input $x_i$.

In order to define a suitable loss-function, we first approximate (assuming independence) the probability that $x_i$ and $x_j$ are assigned to the same cluster for a given $k$ as

$$P_{ij}(k) = \sum_{\ell=1}^{k} P(\ell \mid x_i, k) P(\ell \mid x_j, k).$$

By marginalizing over $k$, we obtain $P_{ij}$ for the probability that $x_i$ and $x_j$ belong to the same cluster:

$$P_{ij} = \sum_{k=1}^{k_{\max}} P(k) \sum_{\ell=1}^{k} P(\ell \mid x_i, k) P(\ell \mid x_j, k).$$

Let $y_{ij} = 1$ if $x_i$ and $x_j$ are from the same cluster (e.g., have the same label) and $0$ otherwise. The loss component for cluster assignments, $L_{\text{ca}}$, is then given by the weighted binary cross entropy as

$$L_{\text{ca}} = \frac{-2}{n(n-1)} \sum_{i<j} \left( \varphi_1 y_{ij} \log(P_{ij}) + \varphi_2 (1 - y_{ij}) \log(1 - P_{ij}) \right)$$

where $\varphi_1 = c\sqrt{1 - \varphi}$ and $\varphi_2 = c\sqrt{\varphi}$, with $\varphi$ the expected value of $y_{ij}$. We use $c$ to normalize the sum of $\varphi_1$ and $\varphi_2$ to 2. Intuitively, we thus account for permutations in the sequence of examples by checking rather for pairwise correctness (probability of same/different cluster) than specific indices.

The second loss term, $L_{\text{cc}}$, penalizes the numbers of clusters and is given by the categorical cross entropy of $P(k)$. The complete loss is given by $L_{\text{tot}} = L_{\text{cc}} + \lambda L_{\text{ca}}$. During training we prepare $N$

batches of size $n$ with $k = 1 \ldots k_{\max}$ clusters chosen uniformly. Note that this training procedure requires only the knowledge of $y_{ij}$ and is thus also possible for weakly labeled data.

## 3 EXPERIMENTAL RESULTS

We evaluate our method on datasets of different modalities recently used in clustering experiments. To compare to related work we measure the performance with the *misclassification rate* (MR) (Liu & Kubala, 2003) and *normalized mutual information* (NMI) (McDaid et al., 2011) .

Table 1: NMI $\in [0, 1]$ and MR $\in [0, 1]$ averaged over 300 evaluations of a trained network.

| | 2D Points (self generated) | | TIMIT | | COIL-100 | |
|---|---|---|---|---|---|---|
| | MR | NMI | MR | NMI | MR | NMI |
| Our method | 0.004 | 0.993 | 0.060 | 0.928 | 0.116 | 0.867 |
| Random cluster assignment | 0.485 | 0.232 | 0.435 | 0.346 | 0.435 | 0.346 |
| Baselines (related work) | k-Means: MR $= 0.178$, NMI $= 0.796$ DBSCAN: MR $= 0.265$, NMI $= 0.676$ | | Lukic et al. (2017): MR $= 0$ | | Yang et al. (2016): NMI $= 0.985$ | |

We conducted evaluations (see Tab. 1) on 2D point data with a high variety of shapes, TIMIT (Garofolo et al., 1993) for speaker clustering and COIL-100 for image clustering. We set $k_{\max} = 5$ and $\lambda = 5$. For the 2D point data we use $n = 72$ network inputs and a batch-size of 200. For TIMIT, the network input consists of $n = 20$ audio snippets with a length of $1.28$ seconds, encoded as mel-spectrograms with $128 \times 128$ pixels (see Lukic et al. (2017) for details). For COIL-100, we use $n = 20$ inputs with a dimension of $128 \times 128 \times 3$. For TIMIT and COIL-100, a simple CNN with 3 conv/max-pooling layers is used as sub-network (a). For TIMIT, we use $430$ of the $630$ available classes for training ($100$ for validation, $100$ for evaluation). We train on COIL-100 using $80$ of the $100$ classes ($10$ for validation, $10$ for evaluation). Example clusterings are shown on Fig. 2.

The results on 2D data as present in Fig. 2a suggest that our method is able to learn specific characteristics of intuitive groupings, giving better results than the traditional methods. Although Lukic et al. (2017) reach better scores for the speaker clustering task and Yang et al. (2016) reach a superior NMI for COIL-100, our method finds reasonable clusterings, is more flexible and can be applied to all data sets. The used code and more details for these and more tests are fully available online[1].

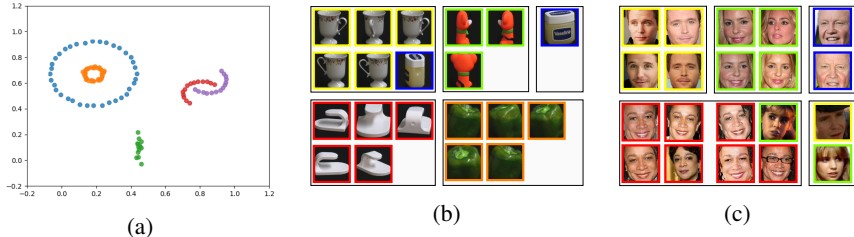

(a)         (b)         (c)

Figure 2: Clustering results for (a) 2D point data, (b) COIL-100 objects, and (c) faces from Face-Scrub (Ng & Winkler, 2014). The colored borders of images depict true cluster membership.

## 4 DISCUSSION & CONCLUSIONS

We have shown that our novel method is able to cluster different data types with promising results. It is a complete end-to-end approach to clustering, that learns both the relevant features and the "algorithm" by which to produce the clustering itself, as well as the number of clusters in the data. The learning phase only requires pairwise labels between examples, and no explicit similarity measure needs to be provided. Promising results are achieved on a variety of modalities.

We observe that the final clustering accuracy depends on the availability of a large number of different classes during training. We attribute this to the fact that the network needs to learn intra-class distances, a task inherently more difficult than just to distinguish between objects of a fixed amount of classes like in classification problems.

---

[1] https://github.com/kutoga/learning2cluster

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
