# OpenReview forum: "Learning to Cluster"
_ICLR.cc/2018/Workshop — Reject_

### Official Review · AnonReviewer3 · 2018-03-02
**This is a supervised method, not clustering.**

**Rating:** 1
**Confidence:** 5

**Review:**

It requires y_ij, the supervised information that two items belong to a cluster or not. This is not cluster analysis.

---

> ### Public Comment · ~Thilo_Stadelmann1 · 2018-03-21
> **Misunderstanding?**
>
> Sorry for not having stated this more clearly in the paper: y_ij is only used during the training phase - the "on-the-fly-creation" of a specific clustering algorithm for a certain class of problems (see my more detailed explanation above). During clustering (i.e., after a neural network has been trained for example to find clusters defined by similar density in point clouds, comparable to DBSCAN), absolutely no supervision is used.
>
> Arguably, it took some supervision (i.e., knowledge of the data and what to expect from it on the side of the developer) to devise any of the classical clustering algorithms. Now that we have these algorithms, their application is unsupervised - this is cluster analysis. This is what any of our trained models do.
>
> Training them is not cluster analysis - it is better compared to creating a new clustering algorithm in the first place plus selecting it because of its appropriatness for the problem at hand (please see my more detailed comment above).
>
> In our approach, we describe both phases, whereas the ultimate goal is always cluster analysis.
>
> I thus beg to differ altogether with this review.

---

### Official Review · AnonReviewer2 · 2018-03-05
**Interesting paper but baselines of empirical comparison is not appropriate**

**Rating:** 6
**Confidence:** 4

**Review:**

This paper proposes an end-to-end clustering method.
The problem of achieving the task of clustering by neural networks is an important topic to the conference, and the proposal is interesting and well described.
However, I have a concern regarding with the experimental setting.
The proposed method uses the information of whether or not a pair of clusters belong to the same cluster, which corresponds to the must-link and cannot-link in the context of semi-supervised clustering.
Hence, to evaluate the empirical performance of the proposed method, it should be compared with semi-supervised clustering methods for fair comparison.

---

> ### Public Comment · ~Thilo_Stadelmann1 · 2018-03-21
> **Thanks for the valuable comments**
>
> Thank you very much for the valuable reference to semi-supervised clustering.We will take this up. As mentioned in a more detailed answer above, I am currently not sure if this would really be the right benchmark.

---

### Official Review · AnonReviewer1 · 2018-03-11
**Well motivated, more experiments and baselines can improve the paper**

**Rating:** 5
**Confidence:** 4

**Review:**

This works proposed a semi-supervised clustering algorithm consisting of 1) embedding network; 2) clustering network; 3) cluster-assignment network; and 4) cluster-count estimating network. The embedding network can use different data modalities as input, such as vectorial data, images and speeches. The supervision is in the form of pairwise constraints indicating whether a sample pair should belong to the same cluster. Experimental results on 2d synthetic data, image and speech data demonstrated the usefulness of the proposed approach.

First, the proposed approach belonged to the category of semi-supervised clustering. Therefore, for fair comparisons, instead of unsupervised learning approaches, the baselines should also be able to make use of pairwise constraints, such as constrained clustering (Wagstaff 2001) and metric learning (Xing 2002).

Second, could the authors provide more details to motivate the use of RBDLSTM for building the clustering network? Did the authors try vanilla feed-forward neural network or CNN as well?

Third, the performance of semi-supervised clustering algorithms heavily depended on the number of pairwise constraints. It seemed the authors used n(n-1)/2 constraints, which indicated all labels of the training samples were known. In real-world, getting the labels (or even pairwise constraints) can be very costly. It would be interesting to demonstrate how the proposed approach performed when the number of pairwise constraints varied from some small number to n(n-1)/2.

---

> ### Public Comment · ~Thilo_Stadelmann1 · 2018-03-21
> **Thanks for valuable comments; let's discuss if semi-supervised clustering is really the correct benchmark**
>
> Thank you very much for the valuable insight. The third point (heavy dependence on the number of pairwise constraints) at first sight seems to correspond with our observation that the method profits from many clusters during training (>> the expected number clusters during application phase).
>
> We will take up an improved motivation for the choice of RBDLSTM cells in a future version (your remark 2) - it can be well motivated, the explanation unfortunately fell prey to space constraints. It is along these lines: We need to compare any sample in the application phase with every other in order to determine if they belong to the same cluster, regardless of the sequence in which they arrive (this is the foundation of being able to find an arbitrary and not fixed number of clusters). A bi-directional recurrent network layer allows for this and has been chosen especially for this reason.
>
> Regarding the first remark: thank you very much for the link to semi-supervised clustering. We haven't been aware of this literature yet. A first glance at https://doi.org/10.1002/wics.1270 ("Semi-supervised clustering methods") however makes me doubt if this is really the right benchmark: In short, training our model is (of course) not part of the cluster analysis - it is better compared to creating a new clustering algorithm in the first place plus selecting it because of its appropriateness for the problem at hand. Instead, semi-supervised clustering as defined in the reference above however uses labels also during the application phase (that we don't).
>
> We will look further into this matter.
>
> (
> A more detailed, preliminary contribution to the discussion on semi-supervised clustering, more appealing to intuition than rigor:
>
> Our presented method operates in two distinct modes.
>
> 1. Training: Here, we train an end-to-end clustering *algorithm* from scratch, based on pairwise labels. For example, we want to have a speaker clustering method. Thus, we feed our general architecture (without any adaptations) any speech segments each spoken by a single speaker, with the information (y_ij) of which segments are uttered by the same speaker. This way, we train an "algorithm" that learns to groups by voice similarity (we could, in principle, use the same raw data together with pairwise information which segments are spoken in the same language, in order to train an algorithm to cluster by language). This phase is definitely supervised learning using weak labels. (Let it be noted that the specific data set used for training does not need to have anything in common with the ones expected later during clustering, besides showing the same "phenomenon" - i.e., containing different voices or  languages by which we want to group, in the example above. Specifically, we do not expect that any of the clusters observed during training will be present during clustering - i.e., we assume that the voices or languages used in training are disjunct from those in the application phase.)
>
> We argue that this phase is similar to the original construction of a clustering algorithm like e.g. DBSCAN, plus the choice of this method by the machine learning practitioner for a specific data set because of the well fitting inductive  bias (see Mitchell, 1997) of the method to the phenomena observed in the data set after exploratory analysis: When I for example get a set of points in 2D space that I want to cluster, I will usually visualize them, see (visually) what kind of groups the points form (if any) and pick a suitable algorithm (k-Means if clusters seem spherical, oriented around central points; DBSCAN if clusters seem more to be defined by their relative point density). Our training phase basically automatizes this manual step of selecting the algorithm with a fitting inductive bias, plus the creation of this algorithm, based on examples of the kinds of clusters we want to find. Supervision of any form is used by our method only in this step (as would be necessary as well in the manual approach).
>
> 2. Application (clustering): Here we use our trained model and apply it to a new data set in which we expect similar groups as we have trained the method to. Let us point out that "similar" here is a very weak assumption: For example, we expect voices in the data if we apply our method trainied to group by voice. We do not expect the same voices used during training. The phase exactly corresponds to the application of any other classical clustering algorithm (the ones mentioned above, or hierarchical clustering): we put a set of data in, and get a complete clustering out (i.e., the number of clusters, plus the cluster membership per input example). Absolutely no supervision is used here, the application phase is purely unsupervised.
> )

---

### Decision · Program_Chairs · 2018-03-20
**ICLR 2018 Workshop Acceptance Decision**

**Decision:**

Reject

**Comment:**

Based on the reviews, this paper has not been accepted for presentation at the ICLR workshop. However, the conversation and updates can continue to appear here on OpenReview.